# Small group size promotes more egalitarian societies as modeled by the hawk-dove game

**Kai-Yin Lin** [ID] ◉, **Jeffrey C. Schank** * ◉

Department of Psychology, University of California, Davis, Davis, CA, United States of America

◉ These authors contributed equally to this work.
* jcschank@ucdavis.edu

**Data Availability Statement:** Data are available without restriction in the GitHub repository (https://github.com/kaiyinlin/spatialESS_groupSize).

**Funding:** The author(s) received no specific funding for this work.

## Abstract

The social organization of groups varies greatly across primate species, ranging from egalitarian to despotic. Moreover, the typical or average size of groups varies greatly across primate species. Yet we know little about how group size affects social organization across primate species. Here we used the hawk-dove game (HDG) to model the evolution of social organization as a function of maximum group size and used the evolved frequency of hawks as a measure of egalitarian/despotism in societies. That is, the lower the frequency of hawks, the more egalitarian a society is, and the higher the frequency of hawks, the more despotic it is. To do this, we built an agent-based model in which agents live in groups and play the HDG with fellow group members to obtain resources to reproduce offspring. Offspring inherit the strategy of their parent (hawk or dove) with a low mutation rate. When groups reach a specified maximum size, they are randomly divided into two groups. We show that the evolved frequency of hawks is dramatically lower for relatively small maximum group sizes than predicted analytically for the HDG. We discuss the relevance of group size for understanding and modeling primate social systems, including the transition from hunter-gather societies to agricultural societies of the Neolithic era. We conclude that group size should be included in our theoretical understanding of the organization of primate social systems.

## Introduction

Primate social systems are diverse, ranging from highly egalitarian to highly despotic. These diverse social systems or societies are characterized by different degrees of cooperative and competitive behaviors. Egalitarian societies are characterized by more cooperative behaviors with less hierarchical organization, while despotic societies are characterized by more competitive behaviors and dominance hierarchies [1]. The distribution of resources also marks the distinction between egalitarian and despotic societies. In more egalitarian societies, the distribution of resources is more equitable, whereas, in more despotic societies, resources are distributed less equitably [1]. This distinction is captured by the classification of primate social systems along the tolerance-conciliation spectrum with four grades based on dominance styles: despotic, tolerant, relaxed, and egalitarian [2]. Despotic species (grade 1) are more

**Competing interests:** The authors have declared that no competing interests exist.

hierarchical, kin-based, and aggression is severe but rarely displayed by subordinates (e.g., rhesus, *M. mulatta*, and Japanese, *M. fuscata*, macaques) [3]. In contrast, egalitarian species (grade 4) are bi-directional with low-intensity aggression, displaying a low degree of asymmetry in agonistic relationships, and are usually loose-kin societies (e.g., Tonkean, *M. tonkeana*, and Moor, *M. maura*, macaques) [4, 5]. Finally, computational models of despotic and egalitarian societies have found that the degree to which a society is egalitarian emerges from the intensity of aggression in conflicts [6, 7].

Human societies also range from despotic to egalitarian, and the degree to which societies are despotic or egalitarian changes over space and time. For most of human evolution, people lived in relatively egalitarian hunter-gather communities [8, 9]. The size of hunter-gatherer communities was relatively small, usually less than 100 [10]. Egalitarianism readily evolved for small groups in hunter-gatherer societies, but as group size increased, societies became more hierarchical and less egalitarian [11]. Dominance hierarchies began to emerge with the start of the agricultural revolution (i.e., the Neolithic revolution) when humans began accumulating resources via the cultivation of food and the size of human groups increased [12, 13]. At that time, extreme violence and homicide rates increased, which were more brutal than today [14]. Thus, the degree of egalitarianism changed over time in human societies, as did the size of groups humans lived in. In particular, group size has mattered in human evolution and, more generally, primate evolution [8, 15–17].

From an evolutionary perspective, theoretical insight into the social organization of societies may be achieved by assessing both the role of the intensity of aggression in conflicts in interindividual conflicts and the role of group size on how egalitarian or despotic a species is. The hawk-dove game (HDG) [18] provides a simple but rigorous model of intra-individual aggression within societies and a model for how resources are distributed [4]. Though highly simplified, the HDG has been used to model egalitarian and despotic societies in addition to animal conflict more generally [19, 20].

In the HDG, individuals must decide how to divide a resource. Individuals playing the hawk strategy fight with other hawks over resources at some cost to the hawk participants. The cost of such conflicts is interpreted here as a measure of the intensity of aggression. If the cost of competition is high, the intensity of aggression is high. If the costs are relatively low, the intensity of aggression is low. Individuals playing the dove strategy do not fight hawks but readily concede a resource with no conflict costs to themselves or their hawk opponents. When doves contest a resource, they share it without conflict or cost. The HDG can be readily solved for evolutionarily stable strategies or mixtures of strategies in a population (see Eq 1 for the HDG payoff matrix). The degree to which a society is egalitarian or despotic depends in this model on the frequency of hawks. The greater the proportion of hawks, the more despotic the society, whereas the lower the proportion of hawks, the more egalitarian a society is.

$$
\begin{array}{c|cc}
 & Hawk & Dove \\
\hline
Hawk & \dfrac{b-c}{2}, \dfrac{b-c}{2} & b, 0 \\
Dove & 0, b & \dfrac{b}{2}, \dfrac{b}{2}
\end{array}
\tag{1}
$$

In the classical HDG, the evolutionarily stable frequency of hawks depends only on the relationship between costs, $c$, and benefits, $b$. If the benefits, $b$, to hawks exceed the costs, $c$, then a population will consist entirely of hawks and is entirely despotic. However, if the costs of hawk-hawk conflicts are high such that $c > b$, then both hawks and doves can stably exist in a population. Indeed, the frequency of hawks in a population can be calculated from

Eq 1 and is $b/c$. Thus, the higher the costs of aggression relative to the benefits, the more egalitarian a given society will be. For example, a society composed of 75% doves would require the cost of aggression to be four times greater than the benefit obtained by fighting and winning. If, as assumed here, the cost of aggressive conflicts between hawks is a measure of the intensity of aggression, then an apparent paradox arises. On an HDG model of despotic and egalitarian societies, societies are increasingly egalitarian due to the increasing intensity of aggression of hawk-hawk conflicts. This result is not consistent with other theoretical results reporting that the degree to which a society is egalitarian emerges from lower levels of intensity in aggressive conflicts [6, 7].

Network and graph studies have emphasized how different spatial structures influence or promote the evolution of cooperation through playing evolutionary games [21–28]. Most studies have focused on the prisoner's dilemma [21, 26, 29], a few on snowdrift (SDG, which has a slightly different payoff matrix than the HDG) [22], or comparing both [30]. Most of them concluded that cooperation thrives on spatial structures with either strong pairwise ties [24], or high viscosity [27, 29], or a specific type of network [31, 32]. A central idea of these network studies is that viscosity or assortment on networks plays a critical role in promoting cooperation [29]. However, one theoretical result indicates that spatial structure inhibits cooperation, particularly in the snowdrift game [22]. In their ABM model [22], specific spatial structures reduce the frequency of cooperators in the SDG when filament-like clusters form while promoting cooperation in PDG when compact clusters form. Although this study [22] concluded that the differences emerge from the game types, these results imply that the cluster pattern and size affect the degree of cooperation.

Prior analytical work on group size indicates that group size affects the frequency of cooperators in threshold public goods games [33–36], which suggests that group size is also important for other cooperative games. Here we theoretically investigated the effects of group size (i.e., clustering size) on egalitarian behavior in group-structured populations with HDG, which is very similar to SDG. To do this, we systematically varied group size and the cost of hawk-hawk conflicts. We developed an agent-based model for group-structured populations to investigate how group size modulates the evolution of hawk frequencies. In this model, agents live in groups and accumulate resources to reproduce by playing the HDG with their fellow agents. Groups are characterized by a maximum size, which, when reached, causes the group to randomly fission into two groups of approximately equal size. A group with only three members fuses with another randomly selected group. Different benefit-cost ratios ($b/c$) were investigated for different group sizes to determine how they both contribute to evolved frequencies of hawks. As we show below, small-group size can dramatically reduce the frequency of hawks and thus increase the degree of egalitarianism in a population. That is, the degree of egalitarianism can be significantly increased without increasing the cost of hawk-hawk conflicts when groups are relatively small. This result aligns with our empirical and theoretical understanding of egalitarian primate societies as typically structured by relatively small groups with lower levels of intensity in aggressive conflicts.

## Model

### 1. Model overview

A group-structured population model was developed to assess the effects of group size on the frequency of hawks relative to the classical predictions of the HDG. Populations consisted of groups of agents that played the HDG for resources to reproduce. Agents accumulated resources over time, which they acquired by playing HDGs with other group members. When an agent acquired sufficient resources to reproduce, $T$, it could produce a single offspring if

the population was below its maximum capacity. When it reproduced, its offspring either dispersed to another randomly selected group in the population with probability $d$ or remained in its parental group with probability $1 - d$. To model only the effect of resource accumulation by playing the HDG fitness (i.e., fecundity only), agents had an average life-span, $L$, with variation among agents, unaffected by the accumulation of resources. The model is graphically depicted in Fig 1. The key properties and functions of agents and groups are described below.

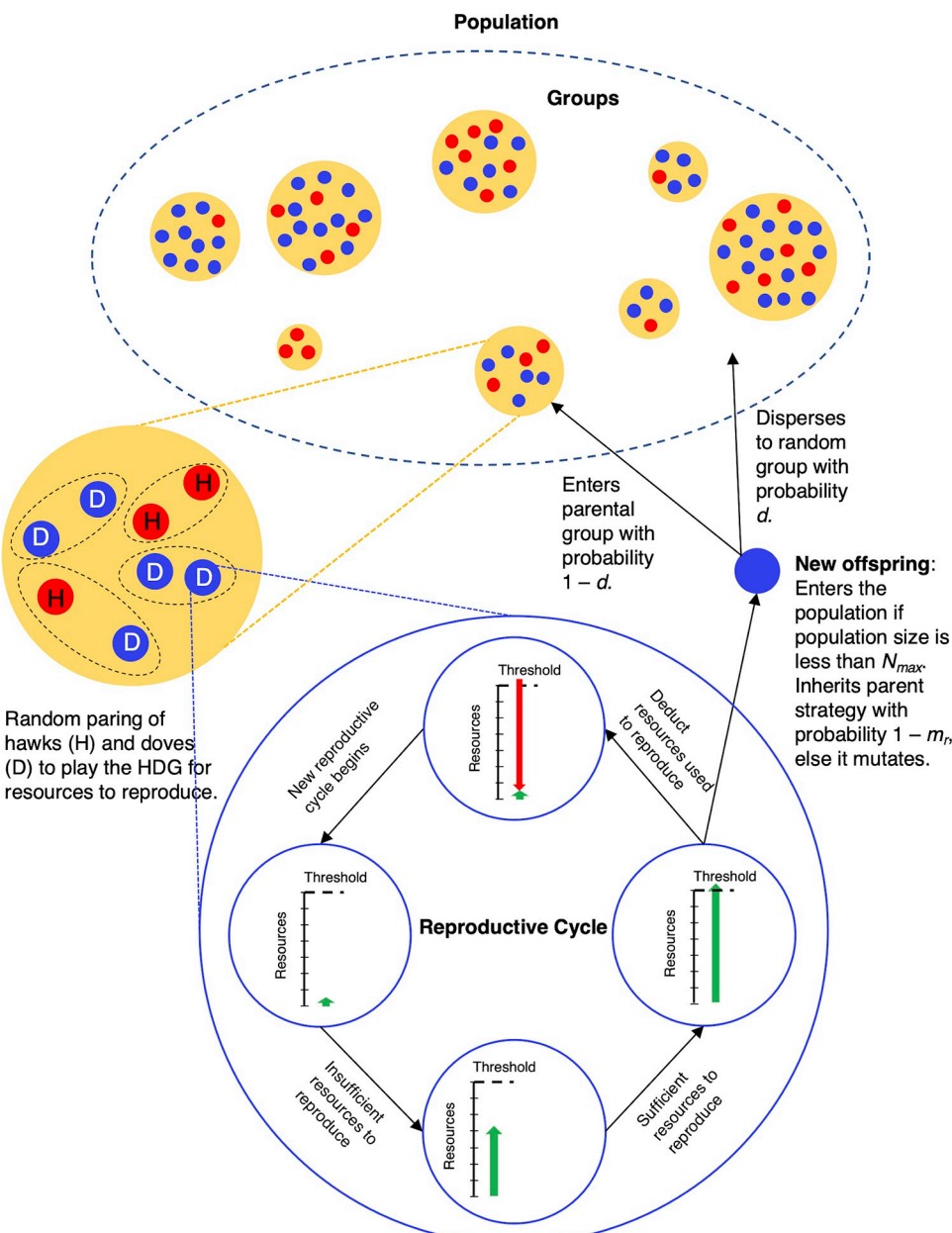

**Fig 1. Overview of the model.** The solid red circles indicate the hawks, and the blue circles indicate doves. Groups are shown in yellow with agents in them. The reproductive process is illustrated by a cycle illustrating the accumulation of resources and then reduction in resources when a reproductive event occurs (bottom).

**Table 1. Fixed parameters, initial conditions, and parameter swept.**

| Parameters | Values | Description | Simulations |
|---|---|---|---|
| Fixed | | | |
| $L$ | 150 | Average lifespan | All |
| $L_{SD}$ | 15 | Lifespan SD | All |
| $T$ | 30 | Resources to reproduce | All |
| $m_r$ | 0.01 | Mutation rate | All |
| $d$ | 0.0 | Dispersal rate | 2 |
| $b$ | 6 | Benefit | All |
| $c$ | 8 | Cost | 3 |
| $GS_{min}$ | 3 | Minimum group size | 2, 3 |
| $N_{max}$ | 10,000 | Maximum # of agents | All |
| Initial conditions | | | |
| $R_0$ | 0 | Starting resources | All |
| $L_0$ | uniform random integer $[0, L]$ | Age | All |
| $N_0$ | 2400 (1200 hawks, 1200 doves) | Population size | All |
| $GS_0$ | $N_0/(GS_{max}/2)$ | Number of groups | 2, 3 |
| Swept | | | |
| $c$ | 6.5, 6.67, 8, 10, 12, 14 | Cost | 1 |
| $c$ | 6, 8, 12 | Cost | 2 |
| $GS_{max}$ | 12, 16, 20, 30, 40, 50, 60, 70, 80 | Maximum group size | 2, 3 |
| $d$ | 0.0, 0.1, 0.2, 0.3, 0.4, 0.5, 0.6, 0.7, 0.8, 0.9, 1.0 | Dispersal rate | 3 |

## 2. Populations

Populations were subdivided into groups with maximum ($GS_{max}$) and minimum ($GS_{min}$) sizes (Table 1). The number of agents in a group changed over time as agents were born, died, or dispersed to other groups. When a group reaches $GS_{max}$, the group fissions into two groups of approximately equal size. When a group drops below $GS_{min}$, it fuses with a randomly selected group from the population. Populations were limited by the total number of agents in the population, $N_{max}$.

## 3. Agents

**3.1 Hawk-dove game.** Agents acquired resources to reproduce by playing HDGs with other agents. Agents were pure hawks or doves; agents did not play mixed strategies. However, a mixture of pure strategies could evolve in a population depending on the benefits ($b$) and costs ($c$) of playing. Their payoffs for all pairwise combinations of strategies are specified in Eq (1). In this study, the benefit was assumed to be less than the cost of a flight according to the classical hawk-dove game. On each time step, an agent played a randomly selected member of its group and played at most one HDG per time-step. This implied that occasionally an agent did not play on a given time step when a group had an odd number of agents.

**3.2 Reproduction.** Agents could reproduce when they accumulated sufficient resources (by repeatedly playing the HDG) to reach a threshold level of resources ($T$). However, to maintain a population at no greater than its specified maximum capacity, $N_{max}$, a Moran [37] like process was implemented such that an offspring agent entered the population only if the population was below $N_{max}$; otherwise, it did not (i.e., it dies). After a reproductive event (i.e., whether or not an agent's offspring successfully entered the population), an agent's resources were reduced by the $T$ resources required to produce an offspring. Offspring mutated to a

different strategy (i.e., hawk → dove or dove → hawk) from their parent with probability $m_r$; otherwise, they inherited their parent's strategy with probability $1 - m_r$. An offspring agent entered its parental group or randomly dispersed to a randomly selected group in the population with probability $d$.

**3.3 Lifespan.** All forms of adult mortality (i.e., predation, disease, etc.) were captured in a lifespan distribution for a population. Using a Gaussian distribution with mean lifespan $L$ and standard deviation $L_{SD}$, values were randomly drawn from a Gaussian distribution for each agent at birth and rounded to the nearest integer for its lifespan. Variation was introduced to avoid synchronous reproduction (i.e., when most of the agents reproduced within one or a few time-steps of each other). The mean lifespan, $L$, was selected so that, on average, agents would have several reproductive events during their lifespan. (See Table 1). In some contexts, the distribution of lifespans (e.g., exponential lifespan distributions) can play a role in maintaining cooperation [38], but here we used a Gaussian lifespan distribution only to avoid synchronous reproduction, which we have observed when there is no variation in agent lifespans.

## Simulations

The model was written in Java using the MASON agent-based library [39]. Each simulation ran for $10^7$-time steps with 10 replicates for each treatment condition. At the start of a simulation, the proportion of hawks to doves in each group was 50%, and the size of each group was $GS_{max}/2$. Each simulation started with 2400 agents equally divided into $2400/(GS_{max}/2)$ groups. The initial age of agents was a uniform random integer in the range 0 to $L$, which distributed agent reproduction out over time. Table 1 provides fixed parameters, initial conditions, and values for parameter sweeps for all the simulations.

Three simulation studies were conducted to investigate how group size affects the evolution of hawk frequencies as the cost of conflict, $c$, was varied. The first set of simulations (1) were validation simulations aimed at establishing that the agent-based model yields the predicted evolutionarily stable hawk frequencies (i.e., $b/c$ when $c > b$) in large mixing populations of agents. Generic biological properties were implemented in these validation simulations, so the prediction of evolutionary game theory can be tested under relatively realistic biological conditions. The second set of simulation studies (2) investigated deviations from the predicted hawk frequencies of $b/c$ when $c$ was varied. Group size effects were systematically investigated in the second set of simulations. The effects of group size were investigated by varying maximum group sizes across simulations. In general, evolutionary stable states were expected to deviate from the analytically predicted evolutionary stables states (calculated from the payoff matrix alone, Eg. 1) as maximum group size changed. The third set of simulation studies (3) investigated the effects of group size when offspring dispersal rate, $d$, was systematically varied. As the dispersal rate, $d$, at birth increased, populations become more mixing. Thus, as dispersal rates increase, hawk frequencies should approach the analytically predicted $b/c$ equilibrium frequency.

## Results

### 1. Validation

With no population structure, any agent could play any other random agent in a population. Under population mixing, all agents evolved to the analytically predicted hawk frequency of $b/c$ when $c > b$ except when $b/c > 0.9$ (Fig 2). When $b/c > 0.9$, doves were eliminated from simulations due to finite population random effects (i.e., selecting agents for games, successful reproductive events, and mutation). When doves were eliminated, because the payoff for

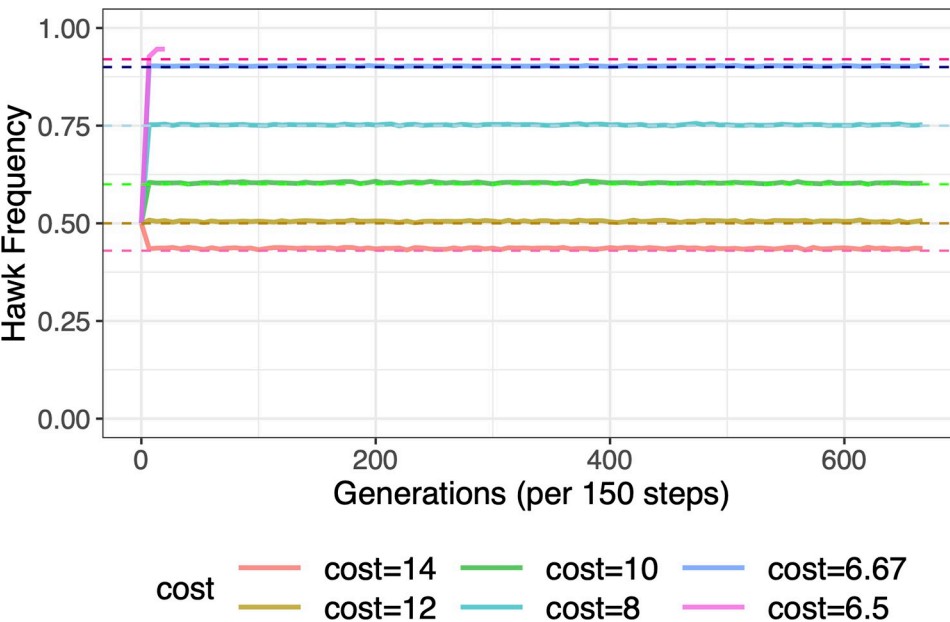

**Fig 2. Validation simulations of theoretical evolutionary stable state (ESS) in hawk-dove game.** The hawk frequencies evolved to the predicted $b/c$ when $c > b$, except when $b/c > 0.9$. Dashed lines indicate the theoretically predicted hawk frequencies, $b/c$. The benefit was held constant $b = 6$ across all simulations. The cost, $c$, was varied, $c = 14, 12, 10, 8, 6.67$, and $6.5$, with predicted hawk frequencies of $b/c = 0.43, 0.5, 0.6, 0.75, 0.9$, and $0.92$, respectively.

hawks was $(b-c)/2 < 0$, population went to extinction. Thus, with the exception of values of $b/c > 0.9$, hawk frequencies evolved to the predicted $b/c$ when $c > b$.

## 2. Group size

When maximum group size was systematically varied (see Table 1), the simulated hawk frequencies were lower than the predicted frequency, $b/c$. For the predicted hawk frequency of $b/c = 6/12 = 0.5$ and small maximum group sizes ($GS_{max} = 12, 16, 20$), the evolved stable hawk frequencies ranged from 10% to 12% (Fig 3A). For larger maximum group sizes ($GS_{max} = 30$ to 80), evolved hawk frequencies increased about 2–4% with each increase of 10 in maximum group size (Fig 3A). In all cases, evolved hawk frequencies were well below the predicted $b/c = 0.5$. For $b/c = 6/8 = 0.75$, evolved hawk frequencies were much lower than the predicted hawk frequency of $b/c = 0.75$ (Fig 3B). For the predicted hawk frequency of $b/c = 6/6 = 1.0$, were at or below 0.5 for group sizes up to $GS_{max} = 30$ (Fig 3C). Interestingly, unlike in the validation simulations with unstructured populations, populations never went extinct even for $GS_{max} = 80$ (Fig 3C).

## 3. Natal dispersal

As the natal dispersal rate increased, evolved hawk frequencies approached the predicted hawk frequency of $b/c = 0.75$, except for smaller maximum group sizes, which overshot the predicted hawk frequency (Fig 4). Interestingly, for small $GS_{max}$ (i.e., 12, 16, 20, 30), dispersal rates greater than 0.5 are required for the frequency of hawks to converge on $b/c = 0.75$. It is also interesting that for small $GS_{max}$, hawk frequencies overshoot $b/c = 0.75$ for dispersal rates over 0.7 (Fig 4). Though not investigated here, these overshoots may have been due to the dispersal of hawks into groups with higher-than-expected frequencies of hawks.

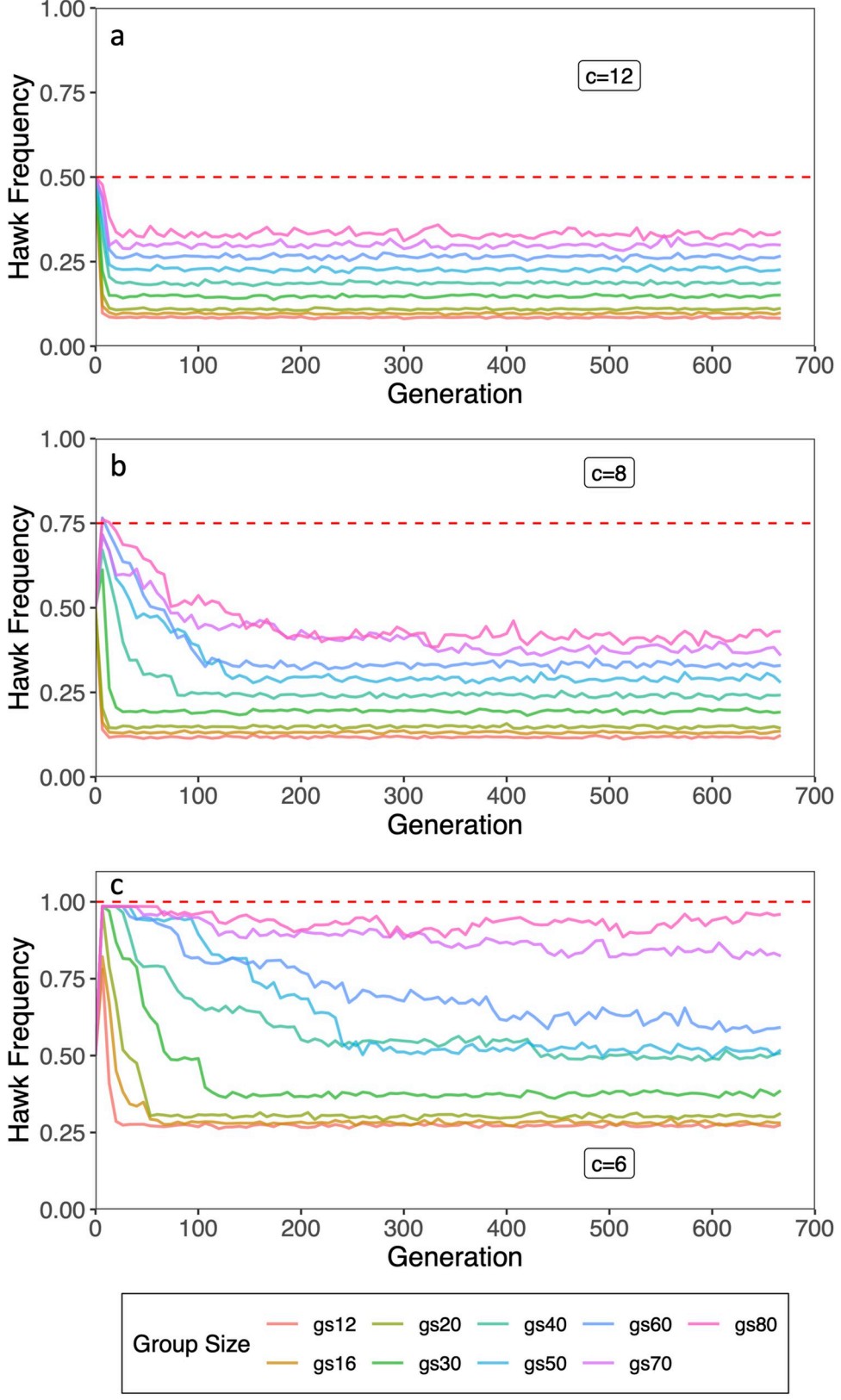

**Fig 3. Evolved hawk frequencies varied by maximum group size and benefit-cost ratio.** Hawk frequencies were lower than theoretical values in simulations with group structure. Dashed lines indicate the theoretically predicted hawk frequencies, $b/c$.

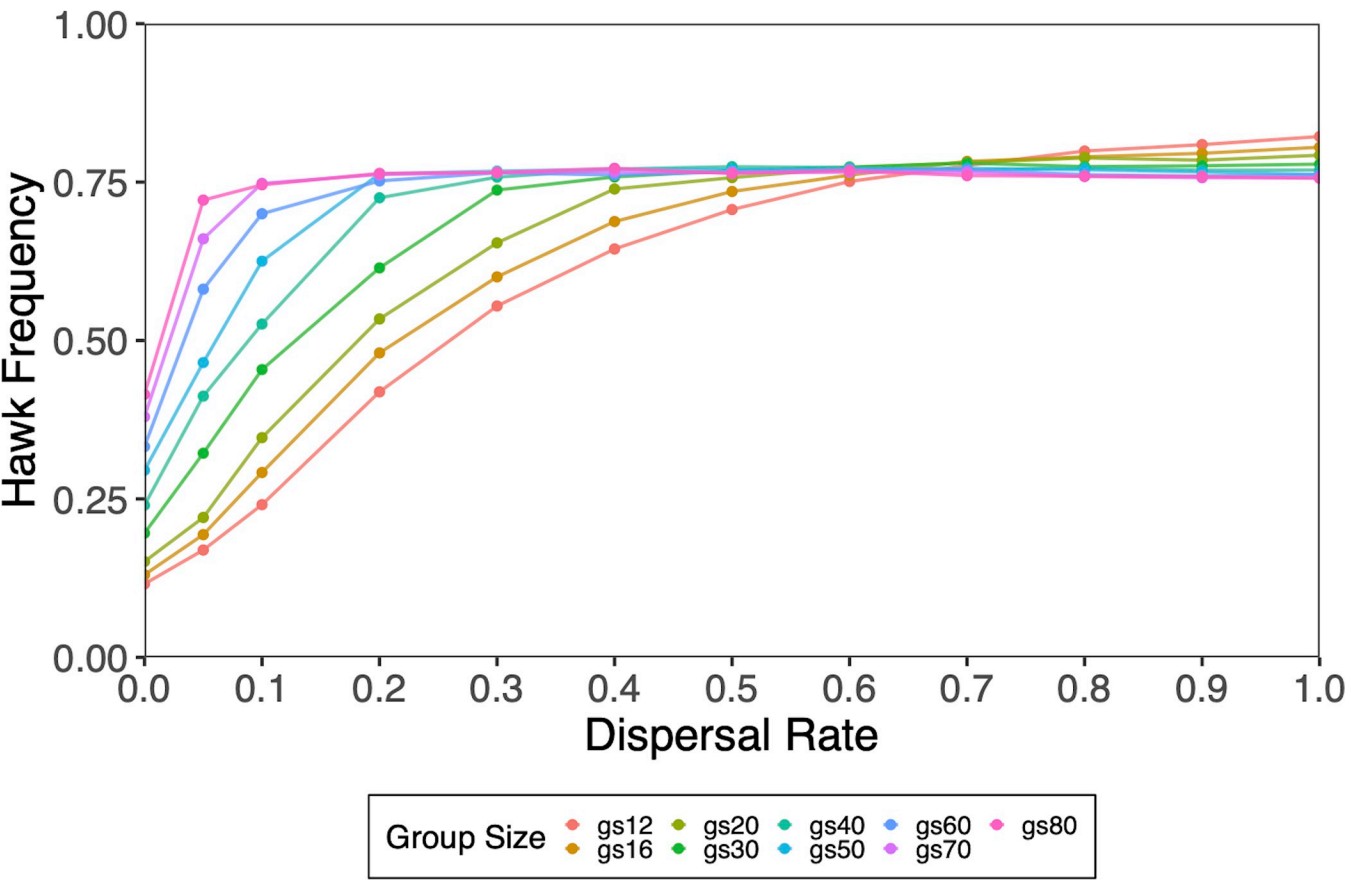

**Fig 4. Natal dispersal for different values of GS$_{max}$ with $b/c = 0.75$.** Dispersal rates of d = 0 to 1.0 were tested. As natal dispersal increases group breaks down; allowing the exploitation of doves by hawks.

## Discussion

Group size resulted in dramatically lower hawk frequencies than predicted by the benefit-to-cost ratio, $b/c$, for the HDG. This was especially true for populations with small $GS_{max}$. Natal dispersal rates reduced dove frequencies by eliminating the effects of group size. These results align with previous work [40], which investigated the impact of mobility radius and probability on cooperation—using Monte Carlo simulations of the prisoner's dilemma and the snowdrift game. They [40] found that either increasing the likelihood of mobility or the radius of probability reduces network reciprocity and cooperation. In our case, the dispersal rate, $d$, can be viewed as one kind of wide-range mobility, which also reduces cooperation. Range and probability of mobility and natal dispersal share a similar effect on population structure: movement breaks down network reciprocity in their can and group structure in ours. Finally, our results demonstrate that small group structure promotes cooperation, which parallels the finding that local movement can promote cooperation [40].

Nevertheless, the effect of small group size on reducing predicted hawk frequencies are robust for high rates of dispersal (Fig 3). Even when half of all offspring born disperse to another random group at birth, hawk frequencies evolved below the predicted frequency of $b/c = 0.75$ for very small group sizes (see Fig 4). These results demonstrate that the size of groups plays a dramatic role in the evolved frequency of hawks. This further suggests that more egalitarian societies could evolve without extremely high costs, $c$, of conflicts if group sizes are relatively small. For example, as illustrated in Fig 4, hawk frequencies evolved to about 10% to 12%

of the population for small group sizes even though the predicted frequency was $b/c = 0.5$. For this case, the cost of conflict, $c$, was twice the benefit, b. However, in the classical HDG, for $b/c$ to fall in the range of 10% to 12% hawks in a population, the cost of hawk conflicts must be 8 to 9 times greater than the benefits for hawk frequencies to be in the range of 10% to 12%. Thus, if the HDG approximately models the range of despotic to egalitarian societies as previously postulated [4, 20], small group size is one way to evolve more egalitarian societies while keeping the intensity of aggression low. Theoretically, these simulation results help reconcile the predictions of the classical HDG that low frequencies of hawks require high costs of conflict with Hemerlrijk's [6, 7, 41] DomWorld model, where low levels of aggression characterize egalitarian societies.

In comparing our results with Hauert and Doebell's SDG model [22], they found that spatial structure filament-like clusters failed to promote cooperative strategies in the snow drift game or HDG. Still, cooperation was promoted in the prisoner's dilemma game with compact cluster formations. Our results demonstrated that small group structure dramatically promoted cooperative strategies (i.e., doves) in HDG. These two seemly opposite results do not conflict. Agents in our model were confined to groups, which are equivalent to complete graphs and more like compact clusters in Hauert's Hauert and Doebell's prisoner's dilemma game model than filament-like structures. For small groups where everyone can interact with everyone else, cooperative strategies will tend to do better. Our results also echo previous network studies that demonstrate cooperation thrives in the networks with strong pairwise ties [24] or structural viscosity [29, 42]. In short, spatial structure does not reduce cooperation but promotes it as long as individuals can form groups.

These results suggest that relatively small social groups or subgroups should characterize more egalitarian primate societies. Egalitarian primate species, such as bonobos, stump-tailed macaques (*Macaca arctoides*), and Tonkean macaques, generally maintain relatively small group or subgroup sizes [43–45]. For example, bonobos live in small groups and subgroups of 2–15 individuals [43]. Bonobos are highly egalitarian, tend not to resolve conflicts through aggression, and are willing to provide food to strangers and express empathy [46]. Even though male bonobos have larger body sizes than females, they do not dominate females as chimpanzees do [44]. In general, bonobos exhibit low aggression intensity and maintain an egalitarian society [44, 47]. Tonkean macaques are also relatively egalitarian as their gradient of dominance is weak [45], and their group size fluctuates between 10–30 individuals over time. Stump-tailed macaques are in the middle of the egalitarian-despotic spectrum and display low aggression with loose hierarchical organization [2]. Their troop size was reported to range between two or three and up to 60 individuals [48], slightly larger than bonobos'. Compared to bonobos, stump-tailed macaques show some hierarchical organization: male individuals sometimes form a strict hierarchy through fighting but can reconcile quickly [49].

In contrast, Japanese macaques and rhesus macaques are both despotic species with higher aggression and less cooperative behaviors. They live in relatively large groups compared to more egalitarian species. The average group size of rhesus macaque is 41.3 [50, 51], but they can range from 22 to 176 [50, 52]. Japanese macaques have an average group size of 40.8 individuals, but they can range to 161 individuals [3].

In the evolution of human societies, early hunter-gatherer groups were largely egalitarian, while more recent human societies have shifted towards more hierarchical organizations [53]. In more egalitarianism hunter-gatherers groups, individuals share food regardless of their ability and generally do not accumulate wealth or property inheritance [54]. The Neolithic era marked the transition from small, egalitarian bands to larger settlements with increased hierarchical organization [9, 53, 55]. Our results have implications for two theoretical approaches to explaining the transition from predominantly egalitarian to hierarchical societies of the

Neolithic era. The first approach emphasizes the important role of resource surplus in decreasing the value of generosity and sharing [56]. In this approach, the hierarchical organization of societies emerges with intra-group conflict over the control of resources. Our results indicate that group size may also play an important role. All things being equal, as the size of groups increases, so does the frequency of hawks and intra-group conflict. The abundance of resources will increase group size, but the increase in group size will contribute to inter-group conflict and thus hierarchy. Another approach focuses on the problem of coordination to achieve cooperative benefits [53, 57–59]. For example, Perret et al. [53] used an evolutionary modeling approach to show that if the cost time in consensus decision-making is great, hierarchical decision-making can evolve in societies even though it results in resource inequalities among individuals. Interestingly, steep hierarchies evolved only as group size became large in their model.

Our results suggest that group size may have played a critical role in the evolution (i.e., biological and cultural) of human cooperation. Both the theories mentioned in the previous paragraph are based on the co-evolution of food surplus and increasing group size. However, the impact of group size in current research is still underestimated. The inevitability formation of leadership hierarchies for large-scale human cooperation is a major focus of anthropological research. Still, such leadership mechanisms have not solved issues of conflict and unfairness in these large-scale cooperative societies.

Our results provide insight into why hunter-gatherers foragers who form small groups are more egalitarian than modern humans living in large communities. Moreover, our results suggest that large-group cooperative issues may at least be partially solvable with subgroup structure. To elaborate on the latter point, modern human societies do not typically have strict linear hierarchies with severe conflict. Today, while societies are hierarchical many maintain some egalitarianism. For instance, corporate structures are hierarchical, often large-scaled, and cooperative. Within hierarchical corporate structures, there are many levels of employees with supervisors and subordinates. However, within a level, team members usually share equal responsibility and power, which is a more egalitarian organization. For example, Super7 uses small teams with a size of seven [60]. Stephen Robbins concluded that teams of more than 12 people had difficulty building trust and function [61]. Individuals in larger teams also perform worse than those in smaller team [62].

Small group size is beneficial for cooperation in many ways. For instance, small group size may be critical to forming egalitarian societies in primates. Small group size provides a natural umbrella for vulnerable cooperators to survive in the population. Empirical research supports the view that egalitarianism co-evolved with small group size in some primate species, hunter-gatherer societies, and perhaps even some modern hierarchical human societies.

In the future, the effect of group size on social systems can be investigated in other evolutionary games to examine if group size is a general mechanism for the biological or cultural evolution of cooperative behavior across different types of cooperative games. Another interesting direction for research would be varying payoffs in the HDG, which would introduce greater biological realism [63].

## Acknowledgments

We thank Matt Miller, Aviva Blonder, and Sydney Wood for comments on earlier drafts of this manuscript.

## Author Contributions

**Conceptualization:** Kai-Yin Lin, Jeffrey C. Schank.

**Data curation:** Kai-Yin Lin.

**Formal analysis:** Kai-Yin Lin.

**Investigation:** Kai-Yin Lin, Jeffrey C. Schank.

**Methodology:** Kai-Yin Lin, Jeffrey C. Schank.

**Project administration:** Jeffrey C. Schank.

**Software:** Kai-Yin Lin.

**Supervision:** Jeffrey C. Schank.

**Validation:** Kai-Yin Lin, Jeffrey C. Schank.

**Visualization:** Kai-Yin Lin, Jeffrey C. Schank.

**Writing – original draft:** Kai-Yin Lin.

**Writing – review & editing:** Jeffrey C. Schank.

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
