## [Decision Letter · Decision Letter 0]

22 Sep 2022

PONE-D-22-23497Small group size promotes more egalitarian societies as modeled by the hawk-dove gamePLOS ONE

Dear Dr. Lin,

Thank you for submitting your manuscript to PLOS ONE. After careful consideration, we feel that it has merit but does not fully meet PLOS ONE’s publication criteria as it currently stands. Therefore, we invite you to submit a revised version of the manuscript that addresses the points raised during the review process.

We look forward to receiving your revised manuscript.

Kind regards,

Luo-Luo Jiang, Ph.D.

Academic Editor

PLOS ONE

Journal Requirements:

3. Please include a caption for Figure 4.

4. Please include a copy of Table 2 which you refer to in your text on page 7.

Reviewers' comments:

Reviewer's Responses to Questions

**Comments to the Author**

1. Is the manuscript technically sound, and do the data support the conclusions?

Reviewer #1: Yes

Reviewer #2: Yes

2. Has the statistical analysis been performed appropriately and rigorously? 

Reviewer #1: Yes

Reviewer #2: Yes

3. Have the authors made all data underlying the findings in their manuscript fully available?

Reviewer #1: Yes

Reviewer #2: Yes

4. Is the manuscript presented in an intelligible fashion and written in standard English?

Reviewer #1: Yes

Reviewer #2: Yes

5. Review Comments to the Author

Reviewer #1: In this paper, the authors model the evolution of social organization by utilizing the hawk-dove game (HDG) and employ the evolutionary frequency of the hawk as a measure of egalitarian/despotism in society. A lower frequency of hawks implies a more egalitarian society, while a higher frequency of hawks indicates a more despotic society. In addition, the authors discuss the significance of group size for understanding and modeling primate social systems. I find that the idea is interesting, and believe their method is valid. Therefore, I can suggest publishing this paper in PLOS ONE after taking the following suggestions into account in a revision with care and love to detail.

1: Figure 1 shows the payoff matrix of the hawk-dove game, which illustrates the payoffs obtained by agents under different strategic interactions. Since it is a matrix, I suggest that it is more appropriate to represent it as an equation rather than a figure.

2: It is advisable for authors to standardize the citation format of figures in the text by using the full name “Figure” or the abbreviation “Fig”.

3: There are some grammatical mismatches and formatting mistakes in the paper. For example, line 207 on page 10 and line 261 on page 13 need to be indented as the beginning of a paragraph. I would suggest the authors proofread the paper meticulously and thoroughly.

4: The authors mention Table 2 in line 135 on page 7, but Table 2 does not appear in the text. It is difficult to give credit to research if even such elementary aspects of the work are not error-free. In addition, I recommend the author not split Table 1 into 2 pages but display it on 1 page, which would be a better layout.

5: The simulations of this paper are not sufficient, and I suggest that the authors could expand them in the following two ways. On the one hand, the authors could elaborate on each simulation in more detail. On the other hand, the authors are generating random numbers to represent the lifespan of agents by using a Gaussian distribution. How would taking other distributions (e.g., exponential distribution, gamma distribution, etc.) affect the results?

6: Many references contain errors and inconsistent formatting. Moreover, most of the authors' references are before 2017 and are too old. The references should be made error-free, and formatted in agreement with the journal guidelines, and references should cite more papers from the past five years. I suggest the authors refer to some of the related papers listed below, which are the most recent papers on games doi: 10.1016/j.physa.2022.126968; 10.1016/j.physa.2022.127297; 10.1063/5.0081954; 10.1063/5.0099444.

Reviewer #2: Small group size promotes more egalitarian societies as modeled by the hawk-dove game

The authors investigated the effects of group size based on Hawk-Dove games, trying to understand the organization in primate social systems. It is very interesting. However, the manuscript (MS) needs to improved in following aspects before I could recommend it to publish in Plos one.

#1. Authors should explain the reason of fixed parameters in table 1.

#2. In section 3.1, authors introduces Hawk-Dove game, as well as the payoff matrix in Fig. 1. However, the important parameters b and c are better to explain in detail in section 3.1 for readers to get a clear image of the game model.

#3. In Fig. 3, authors should set the color bar style as follows: the lower value of in the behind and higher value in the front position. And the color bar are too small to see clearly on the whole.

#4. In Fig. 4, the color bar style are obscure. Author could try to make them bold or bigger to see clearly.

#5. There is too short in main text of fig.4. Are there another aspects to show the effects of group size?

#6. The legends of figures are also simple. Authors may improve their current MS if the figure captions would be made more self-contained. More precisely, what panels are shown for which parameter values, one could also consider a sentence or two saying what is the central theme or message of each figure.

#7. In model section, agents’ offspring either disperses to another randomly selected group in the population with probability d, which is more like the mobility of the agent. Is the parameter d is defined as natal dispersal rate? However, it is very interesting if the authors could discuss the results with the following work for Fig. 5: Li et al.; Applied Mathematics and Computation, 435 (2022) 127456;

#8. Authors investigate the effects of group size theoretically. The work: Jiang et al.; Applied Mathematics and Computation 410 (2021) 126445 explored it experimentally. The section of results could be improved by discussing them.

#9. The authors should check the writing and expression carefully, for example, the referee [22] should be “Nature, 428, (2004) 643–646”. There is a long blank space between line 145 and line 146.

6. PLOS authors have the option to publish the peer review history of their article (what does this mean?). If published, this will include your full peer review and any attached files.

Reviewer #1: No

Reviewer #2: No

---

## [Author Response · Author response to Decision Letter 0]

6 Nov 2022

Reviewer #1: In this paper, the authors model the evolution of social organization by utilizing the hawk-dove game (HDG) and employ the evolutionary frequency of the hawk as a measure of egalitarian/despotism in society. A lower frequency of hawks implies a more egalitarian society, while a higher frequency of hawks indicates a more despotic society. In addition, the authors discuss the significance of group size for understanding and modeling primate social systems. I find that the idea is interesting, and believe their method is valid. Therefore, I can suggest publishing this paper in PLOS ONE after taking the following suggestions into account in a revision with care and love to detail.

1: Figure 1 shows the payoff matrix of the hawk-dove game, which illustrates the payoffs obtained by agents under different strategic interactions. Since it is a matrix, I suggest that it is more appropriate to represent it as an equation rather than a figure.

We changed fig 1 to equation 1 and changed them in the text.

2: It is advisable for authors to standardize the citation format of figures in the text by using the full name “Figure” or the abbreviation “Fig”.

According to the PLOS ONE formatting guideline, we have named Fig both in the text and in the figure title.

3: There are some grammatical mismatches and formatting mistakes in the paper. For example, line 207 on page 10 and line 261 on page 13 need to be indented as the beginning of a paragraph. I would suggest the authors proofread the paper meticulously and thoroughly.

They were both figure titles, so they should not be indented. We also proofread the paper carefully.

4: The authors mention Table 2 in line 135 on page 7, but Table 2 does not appear in the text. It is difficult to give credit to research if even such elementary aspects of the work are not error-free. In addition, I recommend the author not split Table 1 into 2 pages but display it on 1 page, which would be a better layout.

We followed the reviewer’s suggestion to correct the table reference in the text. There should be only one table in this manuscript, so we changed all table 2 to table 1.

We have arranged for table 1 to display on 1 page. Please see the changes on page 9.

5: The simulations of this paper are not sufficient, and I suggest that the authors could expand them in the following two ways. On the one hand, the authors could elaborate on each simulation in more detail. On the other hand, the authors are generating random numbers to represent the lifespan of agents by using a Gaussian distribution. How would taking other distributions (e.g., exponential distribution, gamma distribution, etc.) affect the results?

We elaborated on the simulations in more detail: 

(1) In section 3.1, we elaborate the HDG in more detail. Please see lines 156-163 on page 8.

(2) In section 3.3, we explain why we choose Gaussian distribution for the lifespan simulation. Please see lines 178-187 on page 10. 

(3) In the simulation section, we elaborate three different sets of simulations in more detail. Please see lines 196-211 on page 11-12.

We also explained why we use the Gaussian lifespan distribution. Please see lines 181-187 on page 10. In short, the main reason is that the overall population size in the model can be stable without fluctuations. 

6: Many references contain errors and inconsistent formatting. Moreover, most of the authors' references are before 2017 and are too old. The references should be made error-free, and formatted in agreement with the journal guidelines, and references should cite more papers from the past five years. I suggest the authors refer to some of the related papers listed below, which are the most recent papers on games doi: 10.1016/j.physa.2022.126968; 10.1016/j.physa.2022.127297; 10.1063/5.0081954; 10.1063/5.0099444.

We have checked and followed journal instructions on formatting references. 

 

Reviewer #2: Small group size promotes more egalitarian societies as modeled by the hawk-dove game

The authors investigated the effects of group size based on Hawk-Dove games, trying to understand the organization in primate social systems. It is very interesting. However, the manuscript (MS) needs to improve in following aspects before I could recommend it to publish in Plos one.

#1. Authors should explain the reason of fixed parameters in table 1.

The reasons for fixed parameters were stated in the table 1 caption. More details were explained in section 3.1, 3.2, and 3.3.

#2. In section 3.1, authors introduce Hawk-Dove game, as well as the payoff matrix in Fig. 1. However, the important parameters b and c are better to explain in detail in section 3.1 for readers to get a clear image of the game model.

We also explain the game process in the details to make the game model clear. Please see the simulation section at line 156-163 on page 10.

#3. In Fig. 3, authors should set the color bar style as follows: the lower value of in the behind and higher value in the front position. And the color bar are too small to see clearly on the whole.

Now Fig 3 becomes Fig 2, and we followed the suggestion to re-arrange the legend. The color bars are lengthened and re-ordered as suggested. Please see the file fig2.tif.

#4. In Fig. 4, the color bar style are obscure. Author could try to make them bold or bigger to see clearly.

Now Fig 4 becomes Fig3 because we change Fig 1 to an equation. We have enlarged the whole legend and made the color bars longer to see clearly. 

#5. There is too short in main text of fig.4. Are there another aspects to show the effects of group size?

The main idea in this manuscript is that different group sizes would lead to deviations of the hawk frequencies (i.e., evolutionary stable states) beyond the theoretical b/c. 

#6. The legends of figures are also simple. Authors may improve their current MS if the figure captions would be made more self-contained. More precisely, what panels are shown for which parameter values, one could also consider a sentence or two saying what is the central theme or message of each figure.

We have added the central idea for each figure. Also, the parameters were described in the captions as well. 

Fig. 1 at line 143-146 on page 7

Fig. 2 at line 222-226 on page 12

Fig. 3 at line 241-243 on page 13

Fig. 4 at line 254-256 on page 14

#7. In model section, agents’ offspring either disperses to another randomly selected group in the population with probability d, which is more like the mobility of the agent. Is the parameter d is defined as natal dispersal rate? However, it is very interesting if the authors could discuss the results with the following work for Fig. 5: Li et al.; Applied Mathematics and Computation, 435 (2022) 127456;

In this model, the dispersal rate is defined as the natal dispersal rate, representing a newborn dispersing to another group away from the natal groups. We have discussed the natal dispersal and wide-range mobility effects on cooperative behaviors. In short, both results showed a negative impact on cooperation. The local movement / small group size promotes cooperation, while the wide-range movement/dispersal decreases the cooperation level. Our results align with the suggested paper. We have discussed and cited this article in the discussion section, lines 259-270 on page 14.

#8. Authors investigate the effects of group size theoretically. The work: Jiang et al.; Applied Mathematics and Computation 410 (2021) 126445 explored it experimentally. The section of results could be improved by discussing them.

We cited this paper on page, line 113 in the context of other work on threshold public goods games and group size. 

#9. The authors should check the writing and expression carefully, for example, the referee [22] should be “Nature, 428, (2004) 643–646”. There is a long blank space between line 145 and line 146.

All the reference formats have been checked. 

Journal Requirements:

The repository information can be assessed on git hub, and the link is below:

https://github.com/kaiyinlin/spatialESS_groupSize

3. Please include a caption for Figure 4.

Now Fig 4 becomes Fig3 because we change Fig 1 to an equation. A caption has been added at line241-243 on page 13.

4. Please include a copy of Table 2 which you refer to in your text on page 7.

There is no Table 2 in this manuscript. We have corrected the typo in the main text.

We don’t have supporting information files in this version. The repository information has been uploaded on git hub and we have provided the link above (and in the submission system).

---

## [Editor Report · Decision Letter 1]

12 Dec 2022

Small group size promotes more egalitarian societies as modeled by the hawk-dove game

PONE-D-22-23497R1

Dear Dr. Lin,

We’re pleased to inform you that your manuscript has been judged scientifically suitable for publication and will be formally accepted for publication once it meets all outstanding technical requirements.

Kind regards,

Luo-Luo Jiang, Ph.D.

Academic Editor

PLOS ONE

---

## [Editor Report · Acceptance letter]

15 Dec 2022

PONE-D-22-23497R1 

Small group size promotes more egalitarian societies as modeled by the hawk-dove game 

Dear Dr. Lin:

I'm pleased to inform you that your manuscript has been deemed suitable for publication in PLOS ONE. Congratulations! Your manuscript is now with our production department. 

Kind regards, 

on behalf of

Dr. Luo-Luo Jiang 

Academic Editor

PLOS ONE